# Automated Discovery of Adaptive Attacks
# on Adversarial Defenses

**Chengyuan Yao**                 CHYAO@STUDENT.ETHZ.CH
*ETH Zurich, Department of Computer Science, Switzerland*

**Pavol Bielik**        PAVOL.BIELIK@INF.ETHZ.CH PAVOL@LATTICEFLOW.AI
*LatticeFlow & ETH Zurich, Department of Computer Science, Switzerland*

**Petar Tsankov**      PETAR.TSANKOV@INF.ETHZ.CH PETAR@LATTICEFLOW.AI
*LatticeFlow & ETH Zurich, Department of Computer Science, Switzerland*

**Martin Vechev**               MARTIN.VECHEV@INF.ETHZ.CH
*ETH Zurich, Department of Computer Science, Switzerland*

## Abstract

Reliable evaluation of adversarial defenses is a challenging task, currently limited to an expert who manually crafts attacks that exploit the defenses inner workings, or to approaches based on ensemble of fixed attacks, none of which may be effective for the specific defense at hand. Our key observation is that custom attacks are composed from a set of reusable building blocks, such as fine-tuning relevant attack parameters, network transformations, and custom loss functions. Based on this observation, we present an extensible tool that defines a search space over these reusable building blocks and automatically discovers an effective attack on a given model with an unknown defense by searching over suitable combinations of these blocks. We evaluated our approach on 23 adversarial defenses and showed it outperforms `AutoAttack` (Croce and Hein, 2020), the current state-of-the-art tool for reliable evaluation of adversarial defenses: our discovered attacks are either stronger, producing 3.0%-50.8% additional adversarial examples (10 cases), or are typically 2x faster while enjoying similar adversarial robustness (13 cases).

## 1. Introduction

To address the challenge of adversarial robustness evaluation (Szegedy et al., 2014; Goodfellow et al., 2015), two recent works approach the problem from different perspectives. Tramer et al. (2020) outline an approach for manually crafting adaptive attacks that exploit the weak points of each defense. Here, a domain expert starts with an existing attack, such as `PGD` (Madry et al., 2018) (denoted as ● in Figure 1), and adapts it based on knowledge of the defense's inner workings. This approach was demonstrated to be effective in breaking all of the considered 13 defenses. However, a downside is that it requires substantial manual effort and is limited by the domain knowledge of the expert – for instance, each of the 13 defenses came with an adaptive attack which was insufficient, in retrospect.

At the same time, Croce and Hein (2020) proposed to assess adversarial robustness using an ensemble of four diverse attacks. While these do not require manual effort and have been shown to provide a better robustness estimate for many defenses, the approach is limited by the fact that the attacks are fixed apriori without any knowledge of the particular defense

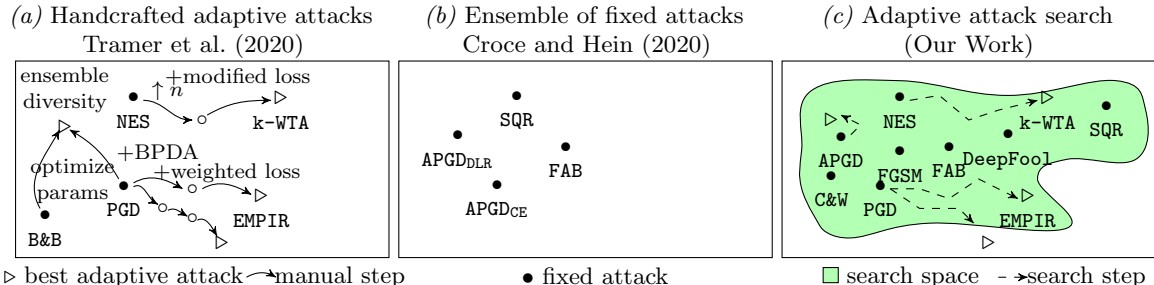

Figure 1: Illustration of recent works and ours. *Adaptive attacks (a)* rely on a human expert to adapt an existing attack to exploit the defense weak points. `AutoAttack (b)` evaluates defenses using an ensemble of diverse attacks. Our work *(c)* defines a search space of adaptive attacks (▢) and performs search steps automatically.

at hand. This is visualized in Figure 1 *(b)* where even though the attacks are designed to be diverse, they cover only a small part of the entire space.

**This work: discovery of adaptive attacks**   We present a new method that automates the process of crafting adaptive attacks, combining the best of both prior approaches – the ability to evaluate defenses automatically while producing attacks tuned for the given defense. Our work is based on the key observation that we can identify common techniques used to build existing adaptive attacks and extract them as reusable building blocks in a common framework. Then, given a new model with an unseen defense, we can discover an effective attack by searching over suitable combinations of these building blocks.

To identify reusable techniques, we analyze existing adaptive attacks and organize their components into three groups: *Attack algorithm and parameters*, *Network transformations*, and *Loss functions*. These components collectively formalize an attack search space induced by their different combinations. We present an algorithm that effectively navigates the search space, and implemented in a tool called `Adaptive AutoAttack` ($A^3$). The source code of $A^3$ and our scripts for reproducing the experiments are available online at:

$$\texttt{https://github.com/eth-sri/adaptive-auto-attack}$$

## 2. Automated Discovery of Adaptive Attacks

We use $\mathcal{D} = \{(x_i, y_i)\}_{i=1}^{N}$ to denote a training dataset where $x \in \mathbb{X}$ is a natural input and $y$ is the label. An adversarial example is a perturbed input $x'$, such that: *(i)* it satisfies an attack criterion $c$, e.g., a $K$-class classification model $f: \mathbb{X} \to \mathbb{R}^K$ predicts a wrong label, and *(ii)* the distance $d(x', x)$ between the adversarial input $x'$ and the input $x$ is below a threshold $\epsilon$ under a distance metric $d$ (e.g., an $L_p$ norm). Formally, this can be written as:

$$\text{ADVERSARIAL ATTACK}: \quad d(x', x) \leq \epsilon \quad \text{such that} \quad c(f, x', x) \tag{1}$$

**Problem Statement**   Given a model $f$ equipped with an unknown set of defenses and a dataset $\mathcal{D}$, our goal is to find an adaptive attack $a \in \mathcal{A}$ that is best at generating adversarial

samples $x'$ according to the attack criterion $c$ and the attack capability $d(x', x) \leq \epsilon$:

$$\max_{a \in \mathcal{A}, \ d(x',x) \leq \epsilon} \mathbb{E}_{(x,y) \sim \mathcal{D}} \quad c(f, x', x) \qquad \text{where} \quad x' = a(x, f) \qquad (2)$$

Here, $\mathcal{A}$ denotes the search space of all possible attacks, where the goal of each attack $a \colon \mathbb{X} \times (\mathbb{X} \to \mathbb{R}^K) \to \mathbb{X}$ is to generate an adversarial sample $x' = a(x, f)$ for a given input $x$ and model $f$. For example, solving this optimization problem with respect to the $L_\infty$ misclassification criterion corresponds to optimizing the number of adversarial examples misclassified by the model.

In our work, we consider an *implementation-knowledge adversary*, who has full access to the model's implementation at inference time (e.g., the model's computational graph). We chose this threat model as it matches our problem setting – given an unseen model implementation, we want to automatically find an adaptive attack that exploits its weak points, but without the need of a domain expert. To solve the optimization problem from Equation 2, we address two key challenges: *Defining a suitable attacks search space $\mathcal{A}$* (Section 3), and *Searching over the space $\mathcal{A}$ efficiently* (Section 4 and Appendix A).

## 3. Adaptive Attacks Search Space

We define the adaptive attack search space to be $\mathcal{A} \colon \mathbb{S} \times \mathbb{T}$, where $\mathbb{S}$ consists of sequences of backbone attacks along with their loss functions, selected from a space of loss functions $\mathbb{L}$, and $\mathbb{T}$ consists of network transformations. Given an input $x$ and a model $f$, the goal of adaptive attack $(s, t) \in \mathbb{S} \times \mathbb{T}$ is to return an adversarial example $x'$ by computing $s(x, t(f)) = x'$. That is, it first transforms the model $f$ by applying the transformation $t(f) = f'$, and then executes the attack $s$ on the surrogate model $f'$. Note that the surrogate model is used only to compute the candidate adversarial example, not to evaluate it.

**Attack Algorithm & Parameters ($\mathbb{S}$)** The attack search space consists of a sequence of adversarial attacks. We formalize the search space with the grammar:

```
(Attack Search Space)
   𝕊 ::=  𝕊; 𝕊 | randomize 𝕊 | EOT 𝕊, n |, repeat 𝕊, n | try 𝕊 for n |
          Attack with params with loss ∈ 𝕃
```

- $\mathbb{S}; \mathbb{S}$: composes two attacks, which are executed independently and return the first adversarial sample in the defined order. That is, given input $x$, the attack $s_1; s_2$ returns $s_1(x)$ if $s_1(x)$ is an adversarial example, and otherwise it returns $s_2(x)$.

- randomize $\mathbb{S}$: enables the attack's randomized components, which correspond to random seed and/or selecting a starting point within $d(x', x) \leq \epsilon$, uniformly at random.

- EOT $\mathbb{S}$, n: uses expectation over transformation, a technique designed to compute gradients for models with randomized components (Athalye et al., 2018).

- repeat $\mathbb{S}$, n: repeats the attack $n$ times (useful only if randomization is enabled).

- try $\mathbb{S}$ for n: executes the attack with a time budget of n seconds.

- `Attack with params with loss` $\in \mathbb{L}$: is a backbone attack `Attack` executed with parameters `params` and loss function `loss`. We provide the full list of parameters, including their ranges and priors in Appendix C.

**Network Transformations ($\mathbb{T}$)**   At a high-level, the network transformation search space $\mathbb{T}$ takes as input a model $f$ and transforms it to another model $f'$, which is easier to attack. To achieve this, the network $f$ can be expressed as a directed acyclic graph, where each vertex denotes an operator (e.g., convolution, residual blocks, etc.), and edges correspond to data dependencies. Note that the computational graph includes both the forward and backward versions of each operation, which can be changed independently of each other. In our work, we include two types of network transformations:

- *Layer Removal*, which removes an operator from the graph. Each operator can be removed if its input and output dimensions are the same, regardless of its functionality.

- *Backward Pass Differentiable Approximation* (`BPDA`) (Athalye et al., 2018), which replaces the backward version of an operator with a differentiable approximation of the function. In our search space we include three different function approximations: *(i)* an identity function, *(ii)* a convolution layer with kernel size 1, and *(iii)* a two-layer convolutional layer with ReLU activation in between. The weights in the latter two cases are learned through approximating the forward function using the test dataset.

**Loss Function ($\mathbb{L}$)**   Selecting the right objective function to optimize is an important design decision for creating strong adaptive attacks. The recent work of Tramer et al. (2020) uses 9 different objective functions to break 13 defenses, showing the importance of this step. We formalize the space of possible loss functions as follows (see Appendix C for details):

```
(Loss Function Search Space)
   𝕃 ::=  targeted Loss, n with Z | untargeted Loss with Z |
          targeted Loss, n – untargeted Loss with Z
   Z ::=  logits | probabilities
Loss ::=  CrossEntropy | HingeLoss | L1 | DLR | LogitMatching
```

## 4. Search Algorithm

Here we briefly describe the main components of our search algorithm. As the search space of attacks S is large, we employ three techniques to improve scalability and attack quality:

- First, to generate a sequence of $m$ attacks, we perform a greedy search – that is, in each step, we find an attack with the best score on the samples not circumvented by any of the previous attacks.

- Second, we use a parameter estimator model $M$ to select the suitable parameters. In our work, we use Tree of Parzen Estimators (Bergstra et al., 2011), but the concrete implementation can vary.

- Third, because evaluating the adversarial attacks can be expensive, and the dataset $\mathcal{D}$ is typically large, we employ successive halving technique (Karnin et al., 2013; Jamieson and Talwalkar, 2016).

We provide the full algorithm, including a more detailed description and time complexity analysis, in Appendix A.

## 5. Evaluation

We now evaluate $A^3$ on 23 models with diverse defenses on CIFAR-10 dataset. The result shows $A^3$ finds stronger or similar attacks than `AutoAttack` for virtually all defenses:

- In 10 cases, the attacks found by $A^3$ are significantly stronger than `AutoAttack`, resulting in 3.0% to 50.8% additional adversarial examples.

- In the other 13 cases, $A^3$'s attacks are typically 2x and up to 5.5x faster while enjoying similar attack quality.

**The $A^3$ tool**  The implementation of $A^3$ is based on `PyTorch` (Paszke et al., 2019), the implementations of `FGSM`, `PGD`, `NES`, and `DeepFool` are based on `FoolBox` (Rauber et al., 2017) version 3.0.0, `C&W` is based on `ART` (Nicolae et al., 2018) version 1.3.0, and the attacks `APGD`, `FAB`, and `SQR` are from Croce and Hein (2020). We use `AutoAttack`'s *rand* version if a defense has a randomization component, and otherwise we use its *standard* version. To allow for a fair comparison, we extended `AutoAttack` with backward pass differential approximation (`BPDA`), so we can run it on defenses with non-differentiable components; without this, all gradient-based attacks would fail.

We instantiate our search algorithm by setting: the attack sequence length $m=3$, the number of trials $k=64$, the initial dataset size $n=100$, and we use a time budget of 0.5 to 3 seconds per sample depending on the model size. The only exception is `A1`, which uses $\epsilon = 0.03$, $m = 8$ and `A10`, which uses time budget of 30 seconds and $m = 1$. We use TPE for parameter estimation, which is implemented as part of the Hyperopt framework (Bergstra et al., 2013). All of the experiments are performed using a single RTX 2080 Ti GPU.

**Comparison to `AutoAttack`**  Our main results, summarized in Table 1, show the robust accuracy (lower is better) and runtime of both `AutoAttack` (`AA`) and $A^3$ over the 23 defenses. Overall, $A^3$ significantly improves upon `AA` or provides similar but faster attacks.

We note that the attacks from `AA` are included in our search space (although without the knowledge of their best parameters and sequence), and so it is expected that $A^3$ performs at least as well as `AA`, provided sufficient exploration time. The only case where the exploration time was not sufficient was for `B14` where our attack is slightly slower (114 min for $A^3$ vs. 107 min for `AA`), yet still achieves the same robust accuracy (5.16% for $A^3$ vs. 5.15% for `AA`). Importantly, $A^3$ often finds better attacks: for 10 defenses, $A^3$ reduces the robust accuracy by 3% to 50% compared to that of `AA`. In what follows, we discuss the results in more detail and highlight important insights.

*Defenses based on Adversarial Training.* Defenses `A2`, `B11`, `B12`, `B16`, `B17` and `B18` are based on variations of adversarial training. $A^3$ obtains very close results while bringing 1.5–5.5× speedups. Closer inspection reveals that `AA` includes two attacks, `FAB` and `SQR`,

Table 1: Comparison of `AutoAttack` (`AA`) and our approach (`A`$^3$) on 23 defenses.

| | | Robust Accuracy (1 - Rerr) | | | Runtime (minutes) | | | Search |
|---|---|---|---|---|---|---|---|---|
| **CIFAR-10**, $l_\infty$, $\epsilon = 4/255$ | | **AA** | **A**$^3$ | $\Delta$ | **AA** | **A**$^3$ | Speed-up | **A**$^3$ |
| A1[*] | Stutz et al. (2020) | 77.64 | **26.87** | -50.77 | 101 | 205 | 0.49× | 659 |
| A2 | Madry et al. (2018) | 44.78 | **44.69** | -0.09 | 25 | 20 | 1.25× | 88 |
| A3[†] | Buckman et al. (2018) | 2.29 | **1.96** | -0.33 | 9 | 7 | 1.29× | 116 |
| A4[†] | Das et al. (2017) + Lee et al. (2018) | 0.59 | **0.11** | -0.48 | 6 | 2 | 3.00× | 40 |
| A5 | Metzen et al. (2017) | 6.17 | **3.04** | -3.13 | 21 | 13 | 1.62× | 80 |
| A6 | Guo et al. (2018) | 22.30 | **12.14** | -10.16 | 19 | 17 | 1.12× | 99 |
| A7[†] | Ensemble of A3, A4, A6 | 4.14 | **3.94** | -0.20 | 28 | 24 | 1.17× | 237 |
| A8 | Papernot et al. (2015) | 2.85 | **2.71** | -0.14 | 4 | 4 | 1.00× | 84 |
| A9 | Xiao et al. (2020) | 19.82 | **11.11** | -8.71 | 49 | 22 | 2.23× | 189 |
| A10 | Xiao et al. (2020)$_{ADV}$ | 64.91 | **17.70** | -47.21 | 157 | 2,280 | 0.07× | 1,548 |
| **CIFAR-10**, $l_\infty$, $\epsilon = 8/255$ | | | | | | | | |
| B11[*] | Wu et al. (2020)$_{RTS}$ | 60.05 | **60.01** | -0.04 | 706 | 255 | 2.77× | 690 |
| B12[*] | Wu et al. (2020)$_{TRADES}$ | **56.16** | 56.18 | 0.02 | 801 | 145 | 5.52× | 677 |
| B13[*] | Zhang and Wang (2019) | **36.74** | 37.11 | 0.37 | 381 | 302 | 1.26× | 726 |
| B14 | Grathwohl et al. (2020) | **5.15** | 5.16 | 0.01 | 107 | 114 | 0.94× | 749 |
| B15 | Xiao et al. (2020)$_{ADV}$ | 5.40 | **2.31** | -3.09 | 95 | 146 | 0.65× | 828 |
| B16 | Wang et al. (2019) | 50.84 | **50.81** | -0.03 | 734 | 372 | 1.97× | 755 |
| B17[*] | Wang et al. (2020) | 50.94 | **50.89** | -0.05 | 742 | 486 | 1.53× | 807 |
| B18[*] | Sehwag et al. (2020) | 57.19 | **57.16** | -0.03 | 671 | 429 | 1.56× | 691 |
| B19[†] | B11 + Defense in A4 | 60.72 | **60.04** | -0.68 | 621 | 210 | 2.96× | 585 |
| B20[†] | B14 + Defense in A4 | 15.27 | **5.24** | -10.03 | 261 | 79 | 3.30× | 746 |
| B21 | B11 + Rand Rotation | 49.53 | **41.99** | -7.54 | 255 | 462 | 0.55× | 900 |
| B22 | B14 + Rand Rotation | 22.29 | **13.45** | -8.84 | 114 | 374 | 0.30× | 1,023 |
| B23 | Hu et al. (2019) | 6.25 | **3.07** | -3.18 | 110 | 56 | 1.96× | 502 |

[*]model available from the authors, [†]model with non-differentiable components.

which are not only expensive but also ineffective on these defenses. `A`$^3$ improves the runtime by excluding them from the generated adaptive attack.

*Obfuscation Defenses.* Defenses `A4`, `A9`, `A10`, `B15`, `B19`, and `B20` are based on gradient obfuscation. `A`$^3$ discovers stronger attacks that reduce the robust accuracy for all defenses by up to 47.21%. Here, removing the obfuscated defenses in `A4`, `B19`, and `B20` provides better gradient estimation for the attacks. Further, the use of more suitable loss functions strengthens the discovered attacks and improves the evaluation results for `A9` and `B15`.

*Randomized Defenses.* For the randomized input defenses `A9`, `B21`, and `B22`, `A`$^3$ discovers attacks that, compared to `AA`'s *rand* version, further reduce robustness by 8.71%, 7.54%, and 8.84%, respectively.

*Detector based Defenses.* For `A1`, `A5`, and `B23` defended with detectors, `A`$^3$ improves over `AA` by reducing the robustness by 50.77%, 3.13%, and 3.18%, respectively. This is because none of the attacks discovered by `A`$^3$ are included in `AA`. Namely, `A`$^3$ found `SQR`$_{DLR}$ and `APGD`$_{Hinge}$ for `A1`, untargeted `FAB` for `A5` (`FAB` in `AA` is targeted), and `PGD`$_{L1}$ for `B23`.

## 6. Conclusion

We presented the first tool that aims to automatically find strong adaptive attacks specifically tailored to a given adversarial defense. Our key insight is that we can identify reusable techniques used in existing attacks and formalize them into a search space. Then, we can phrase the challenge of finding new attacks as an optimization problem of finding the strongest attack over this search space. Our approach is a step towards automating the tedious and time-consuming trial-and-error steps that domain experts perform manually today, allowing them to focus on the creative task of designing new attacks. By doing so, we also immediately provide a more reliable evaluation of new and existing defenses, many of which have been broken only after their proposal because the authors struggled to find an effective attack by manually exploring the vast space of techniques.

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

## Appendix A. Search Algorithm

Here, we describe our search algorithm that optimizes the problem statement from Equation 2. Since we do not have access to the underlying distribution, we approximate Equation 2 using the dataset $\mathcal{D}$ as follows:

$$score(f, a, \mathcal{D}) = \frac{1}{|\mathcal{D}|} \sum_{i=1}^{|\mathcal{D}|} -\lambda l_a + \max_{d(x',x) \leq \epsilon} c(f, a(x, f), x) \tag{3}$$

where $a \in \mathcal{A}$ is an attack, $l_a \in \mathbb{R}^+$ denotes untargeted cross-entropy loss of $a$ on the input, and $\lambda \in \mathbb{R}$ is a hyperparameter. The intuition behind $-\lambda \cdot l_a$ is that it acts as a tie-breaker in case the criterion $c$ alone is not enough to differentiate between multiple attacks. While this is unlikely to happen when evaluating on large datasets, it is quite common when using only a small number of samples. Obtaining good estimates in such cases is especially important for achieving scalability since performing the search directly on the full dataset would be prohibitively slow.

---

**Algorithm 1:** A search algorithm that given a model $f$ with unknown defense, discovers an adaptive attack from the attack search space $\mathcal{A}$ with the best *score*.

---

```
def AdaptiveAttackSearch
```
    **Input:** dataset $\mathcal{D}$, model $f$, attack search space $\mathcal{A} = \mathbb{S} \times \mathbb{T}$, number of trials $k$, initial dataset size $n$, attack sequence length $m$, criterion function $c$, initial parameter estimator model $M$, default attack $\Delta \in \mathbb{S}$

    **Output:** adaptive attack from $a_{[s,t]} \in \mathcal{A}$ achieving the highest *score* on $\mathcal{D}$

**1**    $t \leftarrow \arg\max_{t \in \mathbb{T}} score(f, a_{[\Delta,t]}, \mathcal{D})$      ▷ Search for surrogate model $t$

**2**    $\mathcal{S} \leftarrow \perp$      ▷ Initialize attack to be no attack, which returns the input image

**3**    **for** $j \leftarrow 1{:}m$ **do**      ▷ Run $m$ iterations to get sequence of $m$ attacks

**4**      $\mathcal{D} \leftarrow \mathcal{D} \setminus \{x \mid x \in \mathcal{D} \wedge c(f, a_{[\mathcal{S},t]}(x, f), x)\}$      ▷ Remove non-robust samples

**5**      $\mathcal{D}_{\texttt{sample}} \leftarrow sample(\mathcal{D}, n)$      ▷ Initial dataset with $n$ samples

**6**      $\mathcal{H} \leftarrow \emptyset$

**7**      **for** $i \leftarrow 1{:}k$ **do**      ▷ Select candidate adaptive attacks

**8**        $\theta' \leftarrow \arg\max_{\theta \in \mathbb{S}} P(\theta \mid M)$      ▷ Best unseen parameters according to the model $M$

**9**        $q \leftarrow score(f, a_{[\theta',t]}, \mathcal{D}_{\texttt{sample}})$

**10**       $\mathcal{H} \leftarrow \mathcal{H} \cup \{(\theta', q)\}$

**11**       $M \leftarrow$ update model $M$ with $(\theta', q)$

**12**      $\mathcal{H} \leftarrow$ keep $|\mathcal{H}|/4$ attacks with the best score

**13**      **while** $|\mathcal{H}| > 1$ **and** $\mathcal{D}_{sample} \neq \mathcal{D}$ **do**      ▷ Successive halving (SHA)

**14**       $\mathcal{D}_{\texttt{sample}} \leftarrow \mathcal{D}_{\texttt{sample}} \cup sample(\mathcal{D} \setminus \mathcal{D}_{\texttt{sample}}, |\mathcal{D}_{\texttt{sample}}|)$

**15**       $\mathcal{H} \leftarrow \{(\theta, score(f, a_{[\theta,t]}, \mathcal{D}_{\texttt{sample}})) \mid (\theta, q) \in \mathcal{H}\}$

**16**       $\mathcal{H} \leftarrow$ keep $|\mathcal{H}|/4$ attacks with the best score

**17**      $\mathcal{S} \leftarrow \mathcal{S}$; best attack in $\mathcal{H}$

**18**    **return** $a_{[\mathcal{S},t]}$

---

**Search Algorithm**  We present our search algorithm in Algorithm 1. We start by searching through the space of network transformations $t \in \mathbb{T}$ to find a suitable surrogate model (line 1). This is achieved by taking the default attack $\Delta$ (in our implementation, we set $\Delta$ to $\texttt{APGD}_{\texttt{CE}}$), and then evaluating all locations where $\texttt{BPDA}$ can be used, and subsequently evaluating all layers that can be removed. Even though this step is exhaustive, it takes only a fraction of the runtime in our experiments, and no further optimization was necessary.

Next, we search through the space of attacks $\mathbb{S}$. As this search space is enormous, we employ three techniques to improve scalability and attack quality. First, to generate a sequence of $m$ attacks, we perform a greedy search (lines 3-17). That is, in each step, we find an attack with the best score on the samples not circumvented by any of the previous attacks (line 4). Second, we use a parameter estimator model $M$ to select the suitable parameters (line 8). In our work, we use Tree of Parzen Estimators Bergstra et al. (2011), but the concrete implementation can vary. Once the parameters are selected, they are evaluated using the *score* function (line 9), the result is stored in the trial history $\mathcal{H}$ (line 10), and the estimator is updated (line 11). Third, because evaluating the adversarial attacks can be expensive, and the dataset $\mathcal{D}$ is typically large, we employ successive halving technique Karnin et al. (2013); Jamieson and Talwalkar (2016). Concretely, instead of evaluating all the trials on the full dataset, we start by evaluating them only on a subset of samples $\mathcal{D}_{\texttt{sample}}$ (line 5). Then, we improve the score estimates by iteratively increasing the dataset size (line 14), re-evaluating the scores (line 15), and retaining a quarter of the trials with the best score (line 16). We repeat this process to find a single best attack from $\mathcal{H}$, which is then added to the sequence of attacks $\mathcal{S}$ (line 17).

**$\texttt{A}^3$ Time Complexity**  We give the worst-case time analysis for Algorithm 1. We denote $T_a$ to be the attack time and $T_r$ to be the search time. We will show that under the per sample per attack time limit of $T_c$:

$$T_a \leq m \times N \times T_c \tag{4}$$

$$T_r \leq 2 \times m \times n \times k \times T_c \tag{5}$$

Where $m$, $n$, $k$ are the number of attacks, initial dataset size, number of trials respectively.

In Algorithm 1, only steps on lines 1,4,8,14 are timing critical as they apply the expensive attack algorithms. Since line 4 is essentially applying the attack on all the samples, the runtime of line 4 counts as $T_a$. The runtime of lines 1,8,14 counts as $T_r$.

$T_a$ consists of the time to apply $m$ attacks. The worst-case runtime here is when each of the $m$ attacks use the full time budget $T_c$ on all the samples (denoted as $N$). This gives the bound shown in Equation 4. For $T_r$, we first analyze the time in lines 8 and 14 for a single attack. In line 8, the maximum time to perform $k$ attacks on $n$ samples is: $n \times k \times T_c$. In line 14, the cost of the first iteration is: $\frac{1}{2} n \times k \times T_c$ as there are $k/4$ attacks and $2n$ samples. By design, the cost of SHA iteration is halved for every subsequent iteration, which leads to time for line 14 to be $n \times k \times T_c$. As there are $m$ attacks, the total time bound for lines 8 and 14 is: $2 \times m \times n \times k \times T_c$.

The runtime for line 1 is bounded by $N \times T_{fast}$ as we run single attack on all the samples. Here, we use $T_{fast}$ to denote the maximum runtime of a fast attack that we run at this stage. This step is typically negligible compared to the subsequent search, i.e.,

$N \times T_{fast} \ll 2 \times m \times n \times k \times T_c$. Overall, we can therefore bound the search runtime by considering the lines and 8 and 14, which leads to the bound from Equation 5.

In our evaluation, we use $m = 3, k = 64, n = 100, N = 10000$. Substituting into Equation 5 leads to $T_r \leq 2 \times 3 \times 100 \times 64 \times T_c \leq 4 \times N \times T_c$. This means the total search time is bounded by the time bound of executing a sequence of 4 attacks on the full dataset. Further, $T_r \leq \frac{4}{3} \times m \times N \times T_c$, which means the search time of an attack is bounded by $\frac{4}{3}$ of the allowed runtime to execute the attack.

## Appendix B. Evaluation Metrics Details

We use the following $L_\infty$ criteria in the formulation:

MISCLASSIFICATION
$L_\infty$ ATTACK     $\|x' - x\|_\infty \leq \epsilon$   s.t.   $\hat{f}(x') \neq \hat{f}(x)$

MISCLASSIFICATION
$L_\infty$ ATTACK
WITH DETECTOR $g$     $\|x' - x\|_\infty \leq \epsilon$   s.t.   $\begin{array}{c} \hat{f}(x') \neq \hat{f}(x) \\ g(x') = 1 \end{array}$

For both, we remove the misclassified clean input as a pre-processing step, such that the evaluation is performed only on the subset of correctly classified samples (i.e. $\hat{f}(x) = y$).

**Sequence of Attacks**   Sequence of attacks defined in Section 3 is a way to calculate the per-example worst-case evaluation, and the four attack ensemble in AutoAttack is equivalent to sequence of four attacks [APGD$_{\text{CE}}$, APGD$_{\text{DLR}}$, FAB, SQR]. Algorithm 2 elaborates how the sequence of attacks is evaluated. That is, the attacks are performed in the order they were defined and the first sample $x'$ that satisfies the criterion $c$ is returned.

---

**Algorithm 2:** Sequence of attacks

**def SeqAttack**
    **Input:**  model $f$, data $\boldsymbol{x}$, sequence attacks $\mathcal{S} \subseteq \mathbb{S}$, network transformation $t \in \mathbb{T}$,
            criterion function $c$
1    **for** $\theta \in S$ **do**
2        $x' = a_{[\theta, t]}(x, f)$;
3        **if** $c(f, x', x)$ **then**
4             **return** $x'$
5    **return** $x'$

---

**Evaluation Metric**   Following Stutz et al. (2020), we use the *robust test error (Rerr)* metric to combine the evaluation of defenses with and without detectors. Rerr is defined as:

$$Rerr = \frac{\sum_{n=1}^{N} \max_{d(x',x) \leq \epsilon, g(x')=1} \mathbb{1}_{f(x') \neq y}}{\sum_{n=1}^{N} \max_{d(x',x) \leq \epsilon} \mathbb{1}_{g(x')=1}} \tag{6}$$

where $g \colon \mathbb{X} \to \{0, 1\}$ is a detector that accepts a sample if $g(x') = 1$, and $\mathbb{1}_{f(x') \neq y}$ evaluates to one if $x'$ causes a misprediction and to zero otherwise. The numerator counts the number

of samples that are both accepted and lead to a successful attack (including cases where the original $x$ is incorrect), and the denominator counts the number of samples not rejected by the detector. A defense without a detector (i.e., $g(x') = 1$) reduces Equation 6 to the standard Rerr. Finally, we define *robust accuracy* simply as $1-$ Rerr.

**Robust Test Error (Rerr)** Rerr in Equation 6 has intractable maximization problem in the denominator, so Equation 7 is the empirical equation used to give an upper bound of Rerr. This empirical evaluation is the same as the evaluation in Stutz et al. (2020).

$$Rerr = \frac{\sum_{n=1}^{N} max\{\mathbb{1}_{f(\boldsymbol{x}_n) \neq y_n} g(\boldsymbol{x}_n), \mathbb{1}_{f(\boldsymbol{x}'_n) \neq y_n} g(\boldsymbol{x}'_n)\}}{\sum_{n=1}^{N} max\{g(\boldsymbol{x}_n), g(\boldsymbol{x}'_n)\}} \tag{7}$$

**Detectors** For a network $f$ with a detector $g$, the criterion function $c$ is misclassification with the detectors, and it is applied in line 3 in Algorithm 2. This formulation enables per-example worst-case evaluation for detector defenses.

**Randomized Defenses** If $f$ has randomized component, $f(x_n)$ in Equation 7 means to draw a random sample from the distribution. In the evaluation metrics, we report the mean of adversarial samples evaluated 10 times using $f$.

## Appendix C. Search Space of $\mathbb{S} \times \mathbb{L}$

### C.1 Loss function space $\mathbb{L}$

In Figure 2 we defined the five loss functions we used in the experiments: Cross Entropy (`CE`), HingeLoss (`Hinge`) (Carlini and Wagner, 2017), Difference in logit ratio (`DLR`) (Croce and Hein, 2020), Logit Matching (`LM`). For Hinge, the confidence value $\kappa$ is set to infinity as to encourage stronger adversarial examples, and $\kappa$ can be a loss parameter in future work.

Recall from Section 3 that the loss function search space is defined as:

```
(Loss Function Search Space)
   𝕃 ::=  targeted Loss, n with Z | untargeted Loss with Z |
          targeted Loss, n – untargeted Loss with Z
   Z ::=  logits | probabilities
Loss ::=  CrossEntropy | HingeLoss | L1 | DLR | LogitMatching
```

To refer to different settings, we use the following notation:

- U: for the untargeted loss,

- T: for the targeted loss,

- D: for the targeted − untargeted loss

- L: for using logits, and

- P: for using probs

For example, we use `DLR-U-L` to denote untargeted `DLR` loss with `logits`. The loss space in evaluation is shown in Table 2. Effectively, the search space includes all the

Table 2: Loss functions and their modifiers. ✓ means the loss supports the modifier. `P` means the loss always uses Probability.

| Name | Targeted | Logit/Prob | Loss |
|------|----------|------------|------|
| $\ell_{\texttt{CE}}$ | ✓ | P | $\ell_{\texttt{CrossEntropy}} = -\sum_{i=1}^{K} y_i \log(Z(x)_i)$ |
| $\ell_{\texttt{Hinge}}$ | ✓ | ✓ | $\ell_{\texttt{HingeLoss}} = \max(-Z(x)_y + \max_{i \neq y} Z(x)_i, -\kappa)$ |
| $\ell_{\texttt{L1}}$ | ✓ | ✓ | $\ell_{\texttt{L1}} = -Z(x)_y$ |
| $\ell_{\texttt{DLR}}$ | ✓ | ✓ | $\ell_{\texttt{DLR}} = -\dfrac{Z(x)_y - \max_{i \neq y} Z(x)_i}{Z(x)_{\pi_1} - Z(x)_{\pi_3}}$ |
| $\ell_{\texttt{LogitMatching}}$ | ✓ | ✓ | $\ell_{\texttt{LogitMatching}} = \|Z(x') - Z(x)\|_2^2$ |

Table 3: Generic parameters and loss support for each attack in the search space. For the `loss` column, "-" means the loss is from the library implementation, and ✓ means the attack supports all the loss functions defined in Table 2. In other columns ✓ means the attack supports all the values, and the attack supports only the indicated set of values otherwise.

| ATTACK | RANDOMIZE | EOT | REPEAT | LOSS | TARGETED | LOGIT/PROB |
|--------|-----------|-----|--------|------|----------|------------|
| FGSM | TRUE | $\mathbb{Z}[1, 200]$ | $^*\mathbb{Z}[1, 10000]$ | ✓ | ✓ | ✓ |
| PGD | TRUE | $\mathbb{Z}[1, 40]$ | $\mathbb{Z}[1, 10]$ | ✓ | ✓ | ✓ |
| DeepFool | FALSE | 1 | 1 | ✓ | D | ✓ |
| APGD | TRUE | $\mathbb{Z}[1, 40]$ | $\mathbb{Z}[1, 10]$ | ✓ | ✓ | ✓ |
| C&W | FALSE | 1 | 1 | - | {U, T} | L |
| FAB | TRUE | 1 | $\mathbb{Z}[1, 10]$ | - | {U, T} | L |
| SQR | TRUE | 1 | $\mathbb{Z}[1, 3]$ | ✓ | ✓ | ✓ |
| NES | TRUE | 1 | 1 | ✓ | ✓ | ✓ |

possible combinations expect that the cross-entropy loss supports only probability. Note that although $\ell_{\texttt{DLR}}$ is designed for logits, and $\ell_{\texttt{LogitMatching}}$ is designed for targeted attacks, the search space still makes other possibilities an option (i.e., it is up to the search algorithm to learn which combinations are useful and which are not).

**C.2 Attack Algorithm & Parameters Space $\mathbb{S}$**

Recall the attack space defined in Section 3 as:

$$\mathbb{S} ::= \quad \mathbb{S}; \mathbb{S} \mid \texttt{randomize } \mathbb{S} \mid \texttt{EOT } \mathbb{S}, n \mid \texttt{repeat } \mathbb{S}, n \mid \texttt{try } \mathbb{S} \texttt{ for } n \mid$$
$$\texttt{Attack with params with loss} \in \mathbb{L}$$

`randomize`, `EOT`, `repeat` are the generic parameters, and for `params` are attack specific parameters. The type of every parameter is either integer or float. An integer ranges from $p$

Table 4: List of attack specific parameters. The parameter names correspond to the names in the library implementation

| Attack | Parameter | Range and prior |
|--------|-----------|-----------------|
| NES | step | $\mathbb{Z}[20, 80]$ |
| | rel_stepsize | $^*\mathbb{R}[0.01, 0.1]$ |
| | n_samples | $\mathbb{Z}[400, 4000]$ |
| C&W | confidence | $\mathbb{R}[0, 0.1]$ |
| | max_iter | $\mathbb{Z}[20, 200]$ |
| | binary_search_steps | $\mathbb{Z}[5, 25]$ |
| | learning_rate | $^*\mathbb{R}[0.0001, 0.01]$ |
| | max_halving | $\mathbb{Z}[5, 15]$ |
| | max_doubling | $\mathbb{Z}[5, 15]$ |

| Attack | Parameter | Range and prior |
|--------|-----------|-----------------|
| PGD | step | $\mathbb{Z}[20, 200]$ |
| | rel_stepsize | $^*\mathbb{R}[1/1000, 1]$ |
| APGD | rho | $\mathbb{R}[0.5, 0.9]$ |
| | n_iter | $\mathbb{Z}[20, 500]$ |
| FAB | n_iter | $\mathbb{Z}[10, 200]$ |
| | eta | $\mathbb{R}[1, 1.2]$ |
| | beta | $\mathbb{R}[0.7, 1]$ |
| SQR | n_queries | $\mathbb{Z}[1000, 8000]$ |
| | p_init | $\mathbb{R}[0.5, 0.9]$ |

to $q$ inclusive is denoted as $\mathbb{Z}[p, q]$. A float range from $p$ to $q$ inclusive is denoted as $\mathbb{R}[p, q]$. Besides value range, prior is needed for parameter estimator model (TPE in our case), which is either uniform (default) or log uniform (denoted with $^*$). For example, $^*\mathbb{Z}[1, 100]$ means an integer value ranges from 1 to 100 with log uniform prior; $\mathbb{R}[0.1, 1]$ means a float value ranges from 0.1 to 1 with uniform prior.

Generic parameters and the supported loss for each attack algorithm are defined in Table 3. The algorithm returns a deterministic result if randomize is False, and otherwise the results might differ due to randomization. Randomness can come from either perturbing the initial input or randomness in the attack algorithm. Input perturbation is deterministic if the starting input is the original input or an input with fixed disturbance, and it is randomized if the starting input is chosen uniformly at random within the adversarial capability. For example, the first iteration of FAB uses the original input but the subsequent inputs are randomized (if the randomization is enabled). Attack algorithms like SQR, which is based on random search, has randomness in the algorithm itself. The deterministic version of such randomized algorithms is obtained by fixing the initial random seed.

The definition of randomize for FGSM, PGD, NES, APGD, FAB, DeepFool, C&W is whether to start from the original input or uniformly at random select a point within the adversarial capability. For SQR random means whether to fix the seed. We generally set randomize to be True to allow repeating the attacks for stronger attack strength, yet we set DeepFool and C&W to False as they are minimization attacks designed with the original inputs as the starting inputs.

The attack specific parameters are specified in Table 4, and the ranges are chosen to be representative by setting reasonable upper and lower bounds to include the default values of parameters. Note that DeepFool algorithm uses the loss D to take difference between the predictions of two classes by design (i.e., targeted − untargeted loss). C&W uses the hinge loss, and FAB uses loss similar to DeepFool. For C&W and FAB, we just take the library implementation of the loss (i.e. without our loss function space formulation).

## C.3 Search space conditioned on network property

In our work, properties of network defenses (e.g. randomized, detector, obfuscation) are used to reduce the search space. `EOT` is set to be 1 for deterministic networks. Repeat is set to be 1 for randomized networks, following the practise of `AA` setting repeat to 1 in its *rand* version. Logit Matching is enabled only when detectors are present since the loss is considered as a loss to bypass detectors.

