# OpenReview forum: "Automated Discovery of Adaptive Attacks on Adversarial Defenses"
_ICML.cc/2021/Workshop/AutoML — AutoML@ICML2021 Oral_

### Official Review · Reviewer_UCVw · 2021-06-13
**Comments**

**Rating:** 7
**Confidence:** 4

**Review:**

In the recent past, there has been a lot of interest to propose techniques that evaluate the robustness of models to different kinds of adversaries. Most of these techniques are handcrafted with the help of domain experts to exploit the weaknesses of a certain set of defences. In this paper, the authors propose a method that automatically derives attacks tuned for the given defence. These derived attacks are a composition of a set of reusable building blocks, such as fine-tuning relevant attack parameters, network transformations, and custom loss functions. The proposed approach is a preliminary step towards a direction of designing efficient automated attacks which are dynamically tuned towards defences, as opposed to current practice of handcrafting them.

---

### Official Review · Reviewer_jqHC · 2021-06-17
**Intersting idea for automatically  attacks discovering**

**Rating:** 7
**Confidence:** 4

**Review:**

This paper proposes Adaptive AutoAttack (A^3), an algorithm that automatically discovers attacks given a new model with an unseen defense, which was previously handcrafted by an expert.  This paper defines 3 key components of adaptive attack tasks and defines a search space for each of the components individually and provides a search algorithm to bring these search spaces together.  Overall the paper is well written and all the concepts and algorithms are well defined and clearly described.  Additionally, the results show that A^3 achieves better performance on one of the benchmark and achieves faster results on one of the other benchmark.  Overall, the result is convincing and the approach shows that it is possible to apply AutoML technique to  the filed that was previously believed to require human experts knowledge and gives it new insight into the filed of attack discovery.

---

### Official Review · Reviewer_TJEd · 2021-06-18

**Rating:** 7
**Confidence:** 2

**Review:**

The paper describes an application in the realm of secure systems, which is based on identifying bulding blocks of adversarial attacks.
The main contribution seems to be (at least for my eye) the definition and parametrization of the attack algorithms, including the loss function that needs to be minimized.
The results typically find a better solution than state-of-the-art methods, or at least find a comparable solution more quickly.

I find it hard to judge this paper. I think this is an interesting application, but on the other hand the results are not all that surprising. The main contribution is probably more interesting for domain experts than for machine learning experts. But compared to other applications that I have seen submitted here, I found this to be less straight-forward and more interesting.

Minor comments:
Tramer at al. outlines -> outline (and other similar cases)

---

### Decision · Program_Chairs · 2021-06-21

Accept (Oral)